# Nutrition and Physical Activity in Optimizing Weight Loss and Lean Mass Preservation in the Incretin-Based Medications Era: A Narrative Review

**DOI:** 10.3390/nu18010131

**Published:** 2025-12-31

**Authors:** Luisa Barana, Michelantonio De Fano, Massimiliano Cavallo, Marcello Manco, Deborah Prete, Carmine Giuseppe Fanelli, Francesca Porcellati, Roberto Pippi

**Affiliations:** 1Diabetology and Endocrinology, Degli Infermi New Hospital of Biella, 13875 Ponderano, Italy; luisa.barana@gmail.com; 2Endocrine and Metabolic Sciences Section, Department of Medicine and Surgery, University of Perugia, 06132 Perugia, Italy; michelantonio.defano@gmail.com (M.D.F.); marcellomanco.mm@gmail.com (M.M.); 3Internal Medicine and Vascular Diseases Unit, Azienda Ospedaliera Santa Maria, 05100 Terni, Italy; 4Healthy Lifestyle Institute, Centro Universitario Ricerca Interdipartimentale Attività Motoria (C.U.R.I.A.Mo.), Department of Medicine and Surgery, University of Perugia, Via G. Bambagioni, 19, 06126 Perugia, Italy; carmine.fanelli@unipg.it (C.G.F.); roberto.pippi@unipg.it (R.P.); 5Department of Medicine and Surgery, Perugia University Medical School, 06126 Perugia, Italy; f.porcellati.fp@gmail.com

**Keywords:** obesity, GLP-1, GIP, incretin-based medications, nutritional supplementations, physical activity

## Abstract

Background/Objectives: Incretin-based medications have transformed obesity management by enabling substantial body weight reduction. However, the rapid and pronounced loss of body mass necessitates a comprehensive, multidisciplinary approach incorporating nutritional and physical activity strategies to preserve lean mass, optimize functional outcomes, and prevent long-term complications. This narrative review provides a critical overview of this emerging clinical concern, which is expected to gain increasing relevance in the coming years. Methods: A literature review was conducted up to 31 October 2025, focusing on studies addressing nutritional, physical activity, and adjunctive interventions in adults with obesity treated with incretin-based medications. Results: Incretin-based agents induce significant weight loss, comparable to bariatric surgery, predominantly targeting adipose tissue. Nevertheless, these medications also cause rapid reductions in muscle and bone mass, often accompanied by nutrient deficiencies, which may compromise metabolic health and physical function. Tailored nutritional strategies—including hypocaloric diets enriched in protein and fiber, as well as amino acid, vitamin, and mineral supplementation—are critical to preserve lean mass and support sustained weight maintenance. Concurrently, structured, supervised physical activity, encompassing aerobics, resistance, and strength training, mitigates muscle loss and enhances functional capacity. Emerging pharmacological agents designed to promote adipose tissue reduction while preserving lean mass, as well as interventions targeting gut microbiota modulation, may represent promising adjunctive strategies to optimize long-term outcomes further. Conclusions: While incretin-based medications produce substantial weight loss, their impact on lean mass underscores the necessity of integrating personalized nutrition, supplementation, and structured exercise to preserve muscle, prevent malnutrition, and optimize long-term health and obesity outcomes.

## 1. Introduction

Obesity is a condition characterized by excessive adiposity, with or without abnormal distribution or function of the adipose tissue (AT), posing a “risk to health”. It affects approximately 650 million people globally, and its prevalence is anticipated to increase in the coming years [1].

The etiology of obesity is multifactorial and still incompletely understood. Genetic, environmental, psychological, nutritional, and metabolic factors can induce alterations in the biological mechanisms that maintain the optimal mass, distribution, and function of AT, thereby resulting in obesity. Recently, a Consensus of 58 experts, representing multiple medical specialties and countries, defined “clinical obesity” as a chronic, systemic illness characterized by alterations in the function of organs, the entire individual, or a combination thereof, that potentially causes life-altering and life-threatening complications. In contrast, they defined “preclinical obesity” as a state of excess adiposity with preserved function of tissues and organs but with an elevated risk of developing clinical obesity and various non-communicable diseases like type 2 diabetes mellitus (T2DM), cardiovascular (CV) disease, metabolic dysfunction associated steatotic liver diseases (MASLD), certain types of cancer, and other disorders [2].

A new era in the treatment of obesity has emerged with the development of glucagon-like peptide-1 receptor agonists (GLP-1 RAs). Initially approved exclusively for the glucose control in people living with T2DM, these agents have demonstrated substantial effects on weight loss (WL), prompting growing interest in their use as anti-obesity medications (AOMs).

GLP-1 RAs are thought to induce weight reduction through actions in central and peripheral neuroendocrine systems involved in the regulation of food intake [3]. Centrally, they activate GLP-1 receptors in the nucleus of the solitary tract and the area postrema, thereby enhancing satiety and attenuating food-related reward through serotonergic and dopaminergic pathways. They also stimulate anorexigenic and suppress orexigenic signaling within the hypothalamus and modulate the mesolimbic system, collectively reducing the hedonic drive to eat [4]. Peripherally, GLP-1 RAs influence gastrointestinal (GI) hormone secretion by inhibiting ghrelin, a key orexigenic hormone, and increasing peptide YY and cholecystokinin, which enhance satiety, facilitate digestion, and slow gastric emptying [5].

Through these mechanisms, incretin-based medications promote body weight reduction primarily by targeting AT [6]. As the magnitude of WL was shown to be dose-dependent, higher-dose formulations were developed specifically for obesity management. Liraglutide and semaglutide, respectively, at doses of 3.0 mg daily and 2.4 mg weekly, are now approved AOMs and represent valuable alternatives to bariatric surgery (BS). Evidence has consistently demonstrated their benefits in individuals with obesity, including substantial WL (often exceeding the 10% WL achieved with intensive lifestyle interventions), improved quality of life (QoL), and reductions in obesity-related organ damage [7,8].

Following the success of these therapies, attention has increasingly focused on body-composition changes associated with weight reduction. While AT loss is clinically desirable, preservation of fat-free mass (FFM)—particularly muscle and bone—is critical to avoid sarcopenic obesity (SO), a condition affecting approximately 42–43% of individuals with obesity and often associated with adverse outcomes [9]. Notably, a significant and frequently under-reported decrease in FFM, including skeletal muscle mass (SMM), bone mass, and bone mineral content, has been observed with GLP-1 RAs treatment. This occurs despite preclinical evidence suggesting that these agents may exert beneficial effects on skeletal muscle by mitigating obesity-related muscle atrophy, modulating inflammation, and improving muscle fiber composition, size, and lipid distribution [10,11]. Additionally, other preclinical studies suggest that GLP-1 RAs may promote skeletal muscle remodeling through changes in fiber type and energy substrate utilization [12].

A recent metanalysis of randomized control trials (RCTs) evaluating incretin-based therapies found that approximately 25% of total WL is attributable to lean mass (LM) loss, with no significant change in relative LM, defined as percentage change from baseline [13]. However, the authors noted that available data does not allow a detailed assessment of changes in specific LM components or potential declines in muscle function.

The precise mechanisms underlying LM loss remain largely unexplained, but are likely multifactorial, involving caloric restriction and homeostatic adaptations accompanying WL [14]. GI adverse effects associated with GLP-1 RAs treatment may also contribute. Nausea, vomiting, diarrhea, and constipation are common, particularly during treatment initiation or dose escalation, though they are generally mild, transient, and resolve spontaneously even with ongoing therapy [15].

A substantial pipeline of incretin-based pharmacotherapies is currently in development, aiming to further enhance efficacy and safety profiles [16]. Among these, novel agents capable of simultaneously targeting multiple receptors have attracted considerable interest. The first of these compounds, tirzepatide—approved for both obesity and T2DM—acts as a dual agonist on the GLP-1 receptor and glucose-dependent insulinotropic polypeptide (GIP) receptor. Interestingly, GIP receptors are expressed in brain regions involved in appetite, satiety, food intake, and energy expenditure, and their activation produces effects overlapping with GLP-1 receptor stimulation, aside from direct effects on GI motility [17].

This narrative review aims to provide a thoughtful examination of a developing clinical concern, such as nutrition and physical activity (PA) in optimizing weight loss and lean mass preservation in people using the incretin-based medications, which are likely to gain prominence in the forthcoming years. By evaluating the available literature and current insights, this review seeks to elucidate key aspects of this topic and to consider its potential implications for patient care and clinical practice.

## 2. Methods

In preparation for this review, a search of the MEDLINE (PubMed) was conducted from February to October 2025, considering only papers published in English and limited to those published within the last ten years. The search terms used were “physical activity and anti-obesity medications” OR “nutrition and anti-obesity medications” OR “diet and anti-obesity medications “ OR “supplements and anti-obesity medications” OR “lean mass loss and anti-obesity medications” OR “lean mass preservation and anti-obesity medications”. The electronic search and eligibility of the studies were independently assessed by two authors (C.F., F.P.). Eligible studies included systematic reviews, narrative reviews, meta-analyses, retrospective or prospective observational studies, randomized controlled trials, and clinical practice guidelines involving adults with obesity. Studies were excluded if they involved animal models or participants aged ≤ 18 years, consisted of editorials, letters, case reports, book chapters, or commentaries, or were published in languages other than English, with any disagreements regarding selection resolved through discussion.

## 3. Results

Based on the findings of the studies reviewed, we have divided this section into several arguments as follows.

### 3.1. Evidence from Clinical Trials on GLP-1 RAs and Tirzepatide in Obesity: Effects on Weight Reduction and the Amelioration of Obesity-Related Organ Damage

Liraglutide, semaglutide, and tirzepatide have demonstrated substantial and clinically meaningful WL in adults with obesity across multiple RCTs. In the SCALE Obesity and Prediabetes trial, liraglutide 3.0 mg daily combined with lifestyle intervention resulted in a mean WL of 8.4 ± 7.3 kg over 56 weeks, significantly greater than placebo (2.8 ± 6.5 kg) [18]. Semaglutide 2.4 mg weekly, evaluated in the STEP program, consistently produced 14.9–17.4% WL [19] and in STEP-8 was superior to liraglutide (−15.8% vs. −6.4%) after 68 weeks, with a comparable safety profile [20]. Tirzepatide has shown even greater efficacy. In SURMOUNT-1, tirzepatide 15 mg weekly induced a mean weight reduction of up to −20.9% at 72 weeks in individuals with obesity and without T2DM, compared with −3.1% with placebo [21]. In SURMOUNT-2, participants with obesity and T2DM experienced WL of up to −14.7% [22]. More recently, SURMOUNT-5 demonstrated the superiority of tirzepatide over semaglutide in both WL (20.2% vs. 13.7%) and waist circumference reduction [23].

Beyond WL, semaglutide and tirzepatide exert significant effects on multiple organ systems affected by obesity. Semaglutide 2.4 mg weekly reduced major CV events in individuals with preexisting CV disease (SELECT trial) [24], improved physical function in adult heart failure with preserved ejection fraction (HFpEF) (STEP-HFpEF trial) [25], reduced pain and analgesics use in knee osteoarthritis (STEP-9 trial) [26], and demonstrated histological benefits in metabolic dysfunction-associated steatohepatitis (MASH) (ESSENCE trial) [27], leading to Food and Drug Administration (FDA) approval for the treatment of MASH with moderate-to-advanced fibrosis [28]. Similarly, tirzepatide reduced HF-related events and improved physical performance in adults with HFpEF (SUMMIT trial) [29], decreased the apnea-hypopnea index in obstructive sleep apnea (OSA) (SURMOUNT-OSA trial) [30], leading to FDA approval for moderate-to-severe OSA in adults with obesity [31], and achieved MASH resolution and fibrosis improvement (SYNERGY-NASH trial) [32].

### 3.2. Change in Body Tissue Composition with Incretin-Based Medications

The reduction in SMM and bone mass—quantified as changes in bone mineral density (BMD)—may have critical implications for overall health, underscoring the need for further research and targeted interventions aimed at preserving muscle function and preventing associated declines in both physical performance and metabolic health. It is essential to note that age-related decreases in muscle mass average 0.47% and 0.37% per year in men and women, respectively [33]. Based on this data, it is evident that treatment with incretin-based medications could result in a loss of LM comparable to that typically observed over a decade of a physiological aging.

With respect to liraglutide, Astrup et al. evaluated body composition in 72 individuals with obesity using dual-energy X-ray Absorptiometry (DXA) following 20 weeks of treatment at doses ranging from 1.2 to 3.0 mg, reporting FM reductions of approximately 13–16% alongside LM losses of approximately 1–3% [34]. In contrast, a more recent systematic review encompassing 15 studies with 960 individuals confirmed significant reductions in body weight, FM, and visceral AT compared with placebo, while indicating inconsistent effects on LM [35]. These findings suggest that LM responses to liraglutide may vary depending on study design, duration, and population characteristics.

Concerning semaglutide, DXA data from a subset of participants in the STEP-1 trial revealed that 40% of total WL (6.92 kg out of 17.32 kg) was attributable to LM reduction, corresponding to 13.2% of baseline total LM [36]. Although muscle typically constitutes approximately half of LM, the precise contribution of muscle versus other non-fat tissues, such as the liver, heart, and other organs and tissues, cannot be determined from these data. However, it is reasonable to infer that muscle accounted for at least half of the LM loss, yielding a conservative estimate of ≥10% SMM reduction in the semaglutide group [37]. More reassuring findings emerged from the SEMALEAN trial, in which LM declined initially but subsequently stabilized despite susbstantial WL, alongside reductions in SO prevalence and a significant improvements in handgrip strength [38]. Consistently, a real-world retrospective U.S. study of 94 obese individuals without T2DM receiving semaglutide 2.4 mg, combined with cyanocobalamin, showed that after approximately three months, mean body WL was 4.57%. This was accompanied by a mean reduction of 2.67 kg in total FM, including a 1.17 kg decrease in trunk fat, and a mean loss of 1.43 kg in LM, which accounted for 31% of the total WL. Importantly, the lean-to-FM ratio improved, in line with RCTs [39].

More recent evidence from a DXA sub-study of SURMOUNT-1, involving 160 participants demonstrated that tirzepatide induced reductions in FM and LM of 33.9% and 10.9%, respectively, compared with 5.3% and 2.6% in the placebo group [40]. In another study using magnetic resonance imaging, tirzepatide 15 mg reduced muscle fat infiltration by 0.44% and decrease FFM volume by 0.76 L over 68 weeks, exceeding predictions from the UK Biobank and suggesting favorable fat redistribution despite concurrent muscle loss [33].

An often-overlooked aspect is the impact of incretin-based therapies on bone mass. The effects of GLP-1 RAs have been investigated in a limited number of studies, primarily focusing on obese individuals with T2DM, yielding heterogeneous results [41]. In euglycemic individuals, liraglutide increased procollagen type I *N*-terminal propeptide (P-PINP) levels by 16%, compared with a 2% decrease in the placebo group [42]. Conversely, another study evaluating semaglutide 1.0 mg weekly found no change in P-PINP levels relative to placebo, while observing a significant increase in collagen type I cross-linked *C*-terminal telopeptide (P-CTX) in the semaglutide group. These changes were accompanied by significantly lower BMD at the lumbar spine, total hip, and tibial in the semaglutide-treated participants [43]. More favorable effects on bone may be expected with tirzepatide, given the expression of GIP receptors in one, which contributes to remodeling by regulating both bone formation and resorption [17].

### 3.3. Evidence of Nutritional Interventions, Supplementation, and PA in Obese People Treated with BS

BS represents, to date, the most efficacious therapy for comorbid obesity, representing a paradigm to analyze to make comparisons and considerations in the incretin-based drugs era. BS leads, in fact, to a huge WL that could reach an amount of more than 30% of the pre-operative weight. During the first weeks and within the three months after surgery, the main part of the excessive weight is lost.

Body composition dramatically changes after BS, with a certain grade of FFM loss: minimizing the amount of FFM loss, due to its detrimental consequences, is challenging for clinicians, especially after BS. Haghighat et al. [44] analyzed how different types of BS affect body composition in a period of at least one year after BD: Roux-en-Y-Gastric Bypass (RYGB) reduced FM both in absolute terms and in percentage more than Sleeve Gastrectomy (SG) and gastric bandage (GB), and seems to preserve better FFM over one year of observation. To deeply analyze the FFM loss after BS, Nuijten et al. [45] investigated not only FFM but also LM and SMM determining that more than 8 kg of FFM and LM were lost during the first year after BS and that the first three months in which there is the biggest percentage of loss is crucial to counterbalance this phenomenon that could affect the health of the individual by reducing resting metabolic rate, muscle quality and leading to sarcopenia. Some more recent evidence suggests that muscle strength is not affected by BS, probably because of a huge reduction in lipid infiltration of muscles due to the significant WL [46].

To date, literary data are not significant to determine the correct amount of protein that must be consumed to preserve FFM after BS, and probably the general indication of consuming at least 60 g of protein a day is not sufficient to counterbalance FFM loss [47,48]. Some literary data demonstrate that a consumption of more than 60 g or 1.2 g/kg of proteins could preserve FFM. Considering the majority of FFM loss occurs within the first months after BS, a period in which is particularly difficult for patients who had just undergone BS to consume nutrients generally, and protein specially, one of the main role of bariatric team is to ensure people to introduce leucine-rich proteins (i.e., lean meat, fish, eggs, dairy, legumes) as the preferred macronutrient to choose and, when impossible, to encourage patient to assume protein supplementation. It has been described, instead, that and excessive consume of carbohydrates (>130 gr/day) is associated with worst WL outcomes after BS and a consume of or more than 60 gr/day of fats leads to a significant worsening of FFM loss. Consuming more than 25 g/day of fiber could be protective against WR after BS. Great attention must be paid to micronutrient and vitamin status (i.e., Vitamin D) [49]. Chrononutrition tries to investigate the relationship between the timing of nutrient consumption and state of health or pathology; to date, there is no strong evidence of an implication of the timing of feeding in WR after BS [50].

Structured PA, in a pre-operative setting, could be helpful both in educational terms (leading obese people to the consciousness of the importance of changing lifestyle) and in post-surgical outcomes (i.e., percentage of WL, muscle strength, and cardiorespiratory fitness (CRF) expressed in VO2 max, even if this evidence is not so robust [51]. People undergoing BS must be prioritized to start, as soon as possible, a structured plan of PA to maximize its efficacy in bariatric outcomes [52]. Lacking a definitive guideline for PA in post-BS, emerging evidence of the important role of resistance EXE has been found [53]. Independent of the characteristics of PA after BS, it could enhance weight loss, muscle strength, and CRF [54,55,56]. People undergoing BS must be engaged in structured PA programs to benefit better of its effects on their health [57].

BS paradigm of weight loss suggests the seminal importance of combining dietary interventions and structured PA to counterbalance lean mass loss during the weight loss process.

### 3.4. Evidence of Nutritional Interventions or Supplementation in Obese People Treated with Incretin-Based Medications

GLP-1 RAs act by slowing gastric emptying and modulating hunger centers, leading to progressive weight loss (especially lean mass, ranging from 20% to 50%) and increasing the risk of malnutrition [58,59,60,61]. GI side effects (e.g., nausea, vomiting, constipation, and diarrhea) may further compromise nutrient and fluid intake and/or absorption, exacerbating malnutrition [58,59,60,61]. Thus, when prescribing an AOM, clinicians should consider not only nutritional aspects, particularly concerning diet quality, but also dietary management of AOM’s side effects (Figure 1). Diet quality enables greater and sustained WL in the long term, whereas side effects can affect metabolic health [62].

*Strategies to face AOMs’ side effects and suggested dietary patterns.* The adverse effects of GLP-1 RAs and GLP-1/GIP Ras encompass a range of symptoms, including nausea, vomiting, diarrhea, and abdominal discomfort, which may vary in severity and duration. They represent a notable concern potentially impacting treatment adherence, efficacy, and success. Moreover, they can alter eating patterns and contribute to nutritional imbalance, a condition often obscured by excess body weight. Symptoms such as reduced appetite, early satiety, and nausea may limit dietary variety and protein intake, leading to muscle loss [63,64,65].

Smaller, more frequent meals and avoiding fatty or high-fiber foods during the first few days of treatment can help alleviate nausea and similar symptoms. Dietary fat naturally slows digestion, and combining a higher-fat diet with GLP-1RA may further increase gastrointestinal discomfort.

Adequate fluid intake, as well as soluble and insoluble fiber from foods, and avoiding protein- or fat-dense foods, should be encouraged to combat constipation (the most common side effects of GLP-1RA treatments) [66]. However, additional strategies are often required, such as the use of magnesium citrate. It is also important to stay well hydrated (more than 2–3 L per day) to protect kidney function, adjusting fluid intake to the type of PA performed [67]. In contrast, avoiding high-fat meals can be helpful for diarrhea [68].

In the presence of gastrointestinal symptoms or a marked reduction in caloric intake, whole-grain products should be limited or avoided. The high fiber content of these foods may exacerbate early satiety—further decreasing meal volume—and can contribute to impaired nutrient absorption. In this context, vegetarian or vegan diets, known for their wide range of health benefits, such as reduction in body weight, cholesterol, and incidence of T2DM [69,70], should be carefully planned. In fact, due to their high fiber content (which can exacerbate an imbalance in micronutrient intake) and low availability of certain vitamins [71] and proteins, it could be helpful to supplement, if necessary, micronutrients and plant-based protein powder to ensure the recommended amount [72].

AOMs, especially GLP-1 RAs and GLP1/GIP RAs, decrease caloric intake by 16–39% [73] and can expose patients to insufficient or inadequate intake of macro- and/or micronutrients, reducing baseline expenditure requirements, mainly when an unbalanced diet occurs, and nutritional counseling is not practiced in clinical routine. A cross-sectional study of 99 participants shows that AOMs users consume an unbalanced diet, with excessive calories from saturated fatty acids, inadequate calories from carbohydrates, and a lack of micronutrients, especially vitamin D [66].

Currently, there is a lack of a comprehensive, evidence-based review specifically addressing the nutritional needs of patients treated with the most recent and effective AOMs [74]. In fact, studies refer to general Dietary Guidelines [62,68,73]. Undoubtedly, if these treatments substantially decrease the quantity of food consumed, it is essential to consider the quality of the food that patients consume. The ability of these medications to selectively reduce the intake of high-fat, high-sugar palatable foods [60,61,75] makes them strategic in nutritional re-education. Along with nutritional counseling, long-term, low-intensity, structured programs that include support for changing food choices and PA enable the maintenance of healthy eating habits after a hypothetical discontinuation of the AOMs [68].

Baseline energy requirements vary according to individual characteristics such as age, sex, body weight, and PA level and should be personalized. Regarding energy distribution and nutrient quality, the diet should adhere to the general guidelines for T2DM and/or obesity (Table 1). These findings suggest limiting the consumption of saturated fat, sugar, and processed foods and promoting the consumption of fruits, vegetables, fiber, and lean proteins, which are well known for their beneficial effects on metabolic health [76]. If T2DM and obesity occur, the Evidence-based European recommendations for the dietary management of T2DM suggest a wide range of carbohydrate intakes, preferring whole cereals rich in fiber (at least 35 g/day); plant-based, both mono- and polyunsaturated fats that cover less than 10% of total energy requirements; and a protein intake of 23–32% in the context of a WL diet.

The Mediterranean diet and the Dietary Approaches to Stop Hypertension diet (MD and DASH, respectively) are the most extensively studied dietary patterns for WL. Notably, the MD has been particularly well-studied due to its efficacy in CV protection [69,77,78,79] and its long-term adherence/sustainability.

In the CARDIVEG Study (Cardiovascular Prevention With Vegetarian Diet) [71], a randomized, open, crossover dietary trial, a vegetarian diet was compared to an MD in healthy overweight adults. Both were effective in promoting weight reduction; however, the vegetarian diet resulted in a greater reduction in low-density lipoprotein cholesterol, whereas the MD led to a more pronounced decrease in triglyceride levels. Notably, adherence to the vegetarian diet was associated with reduced vitamin B12 levels.

An extensive study involving 811 patients examined the WL and metabolic benefits of four dietary patterns, which vary in their proportions of fat, protein, and carbohydrates, in individuals with overweight or obesity after 2 years. Although there was no difference in WL among the four groups, the low-fat and highest-carbohydrate diets decreased LDL cholesterol levels more than the other two. In contrast, the lowest-carbohydrate diet increased HDL cholesterol levels more than the highest-carbohydrate diet [73]. Additionally, adherence was easier for patients who followed a nutritional pattern in accordance with their values and habits.

High-protein hypocaloric diet did not affect changes in FFM during weight loss in overweight and obese older adults in a 10-week weight loss program with or without a resistance exercise program [80]. Only the group with the combined intervention of the high protein diet and the resistance exercise program significantly increased in FFM.

A complete, well-balanced diet, such as the MD, ensures adequate macro-and micronutrient intake and is a highly effective and attainable long-term strategy. Moreover, recommendations suggest Mediterranean, Nordic, or Vegetarian patterns and advise against the use of the ketogenic diet (KD) for WL in patients with T2DM [76]. When people with diabetes are treated with AOMs, the combination of KD could increase the risk of diabetic ketoacidosis [68]. For people on KD, nutritional adequacy needs to be ensured carefully.

In the context of treatment with AOMs, fiber needs are controversial, as a reduction in energy intake may lead to lower fiber consumption than the dietary recommended levels (21–25 g/day for women and 30–38 g/day for men). The POUNDS Lost study [81], which involved 810 participants with obesity (excluding T2DM), demonstrated that a greater increase in fiber intake (>20 g) was associated with better diet adherence and greater WL (~4 kg more at 6 months) compared to those consuming less fiber. Nonetheless, high-fiber consumption during treatment with an AOM can exacerbate GI side effects.

*Protein requirements, Micronutrients, and supplements in Obese People Treated with AOMs.* A significant reduction in daily calorie intake, particularly when combined with an AOM, could permit a significant initial drop in total WL, but also a substantial loss in lean body mass, especially in obese patients with heart failure [58]. Therefore, it might affect sustainable WL and body composition improvements.

Loss of SMM may have clinical and nutritional implications, including impaired metabolic health, increased risk of weight regain (‘yo-yo effect’), and a decline in overall QoL [82]. A critical factor, alongside resistance exercise, for preserving muscle mass during WL is adequate nutrition (with a caloric deficit) [62,68], particularly focused on the intake of proteins and specific micronutrients by the use of dietary supplements [68,74]. According to the American College of Sports Medicine, people with obesity undergoing WL treatment should consume approximately 1.25–1.5 times the Recommended Dietary Allowance (RDA) for daily protein intake (for athletes > 1.5), with at least ∼25–30 g of protein per meal [83,84] to limit muscle mass loss. The United States Department of Agriculture guidelines also recommend an intake of 60–75 g/day of protein, or 1.5 g/kg body weight/day, during body WL treatment [74]. However, a protein intake of 1.5 g/kg/FFM per day is considered more adequate [68]. A strategy to mitigate the loss of LM caused by GLP-1 RAs treatment may involve protein supplementation (particularly with whey or casein proteins), preferably consumed post-exercise [85]. Furthermore, under conditions of calorie restriction, it appears to be beneficial to supplement the daily protein intake with Essential Amino Acids (EAAS) [86], especially to enhance the post-exercise anabolic effect, which would otherwise not be stimulated by the deficit itself [87]. Daily consumption of protein shakes can help meet these requirements [62]. Recently, it has been shown that plant-based proteins can stimulate muscle protein synthesis [88]. The quality of proteins, determined by the content of essential amino acids and digestibility (biological value), affects their ability to support or hinder muscle protein synthesis [89]. According to recent studies, women with sarcopenia who take a high-protein supplement containing Branched-Chain Amino Acids (BCAAs), markedly with L-Leucine, can maintain lean body mass, but there are no differences in muscle strength compared to the placebo group [84,90,91]. Beyond macronutrient adeguacy, hypocaloric dietary strategies implemented during pharmacological therapy may increase the risk of micronutrients deficiencies, which are critical for maintaining muscle mass and function [82]. The risk is further amplified by the appetite suppression commonly observed during AOM treatment [68,74].

Importantly, individuals starting AOM may already present with pre-existing nutritional deficiencies as a consequence of long-standing unbalanced dietary habits associated with obesity.

In particular, deficiencies of water-soluble vitamins, magnesium, and vitamin D are common in individuals with obesity [60]. Magnesium deficiency is frequently associated with unhealthy dietary patterns, such as Western diet, rich in ultra-processed food and low in fruits and vegetables; moreover, simple and refined sugars activate hepatic glucose catabolic pathways, which are magnesium-dependent. Consequent reduction in magnesium in the liver increases NAPDH levels and promotes triglycerides and very low-density lipoprotein synthesis, facilitating obesity [91]. It should be noted that magnesium is also essential for vitamin D synthesis and activation. Vitamin D and magnesium inadequacy are associated with an increased risk of cardiometabolic disease in obese non diabetic subjects, whereas regular levels improve cardiometabolic risk indicators [92]. Iron deficiency is a common finding in obesity, and the low-grade or chronic inflammatory component is considered one of the potential mechanisms of hypoferremia: it impairs iron absorption and metabolism and promotes the sequestration of iron in the adipose tissue [93,94].

Zinc has anti-oxidative stress and anti-inflammatory functions, and a lack of zinc is related to insulin resistance, hyperleptinemia, and chronic low-grade inflammation [95]. The association between zinc deficiency and the development of cardiovascular diseases (CVDs) has been supported by numerous studies [96].

According to recent studies, following AOM treatment, individuals often exhibit inadequate intake or deficiencies of several micronutrients such as retinol, vitamin E, vitamin C, magnesium, B1, B12, iron, zinc, and especially vitamin D, calcium, and B9 [66,68,74]. Changes in fat mass can significantly affect the levels of fat-soluble vitamins, such as vitamin D, because they depend on fat stores [97]. Moreover, side effects, reduced appetite, change in taste, and early satiety may affect micronutrient imbalance and contribute to deficiencies. For example, during AOMs treatment, magnesium depletion can be exacerbated by continued and marked vomiting and/or diarrhea [98].

After 12 months of treatment with GLP-1RA, nutritional deficiencies worsen in 20% of cases, especially regarding vitamin D [59]. Both calcium and vitamin D should be regularly assessed in order to reduce osteoporosis and fracture risk [99]. Adequate vitamin D status is critical not only for bone metabolism but also for skeletal muscle function, maybe influencing muscle protein synthesis, strength, and physical performance [100].

Therefore, in the absence of official guidelines [66], an accurate physical examination and regular blood tests to assess vitamin levels and supplementation with a specific and appropriate multivitamin may be helpful in patients receiving AOMs [62] in proper and personalized doses for each person [68].

Supplements are concentrated sources of nutrients or other substances with a nutritional and physiological effect that are consumed to complement the diet and improve nutritional status. They may contain minerals, vitamins, amino acids, metabolites, herbs, or plant extracts in the form of tablets, capsules, powders, or liquids [101]. Current studies are investigating the use of oral nutritional supplements to preserve muscle mass in individuals with obesity [102] (Table 1). Compounds derived from amino acids, such as taurine (an aminoethanesulforic acid synthetised from cysteine and methionine), glutathione (a tripeptide synthetised from cysteine, glycine and glutamic acid), betaine (derived from amin oacid glicine), α-ketoglutarate (derived from amino acid glutamate), β-aminoisobutyric acid (non-proteinogenic amino acid derivated from valine), and β-hydroxy-β-methylbutyrate (metabolite of amino acid leucine), have been shown to have favorable effects on body weight control and improvement of the lipid profile in animal studies, thus becoming potential candidates for the development of new therapies for obesity. The mechanisms underlying the effects of these amino acid derivatives primarily involve the inhibition of adipogenesis, stimulation of lipolysis, promotion of brown and beige adipose tissue development, and enhancement of glucose metabolism [103]. Numerous dietary supplements have been used to support WL and the treatment of obesity, including β-glucans, glucomannan, guar gum, calcium, vitamin D, chromium, bitter orange, and Garcinia cambogia. Significant WL has been demonstrated with the use of green coffee, green tea, α-Lipoic Acid, cocoa, and chitosan [102,104,105,106]. Supplementation with chromium picolinate, combined with picolinic acid to enhance absorption, appears to be effective in preserving lean body mass and stimulating FM loss [104].

Micro- and macronutrient intake to support WL and preserve LM in adults with obesity is available in Appendix A.

*Microbiota during weight loss: an ace in the hole*? The role of gut microbiota in obesity is still controversial. It is well known that gut microbiota mirrors the metabolic status of the individual. Exposure to dietary patterns rich in fats or processed foods changes gut microbiota. Lean individuals present an extremely different microbiota status than obese people.

Since obesity is a complex and multifactorial condition influenced by modern lifestyle, alterations in the gut microbiota, epigenetic, and hormonal factors [107], there has been growing interest in the use of postbiotics (metabolites produced by probiotics) as a potential therapeutic approach [108]. Evidence from the literature suggests that the ratio of Firmicutes to Bacteroidetes dramatically changes according to the weight of the individual. The bioproducts elaborated by Firmicutes (i.e., SCFA, LPS) explain the pro-inflammatory status typically present in obese people. Although it has been shown that gut metabolites influence incretin secretion, further evidence is needed to determine whether changes in the microbiota are a cause of obesity or a consequence of it [109].

Little is known about the influence of AOMs on microbiota. G LP-1 receptor agonists (GLP1RAs) increased the intestinal Bacteroidetes to Firmicutes ratio, decreased obesity-related but increased lean-related microbiota phenotypes, and implement bacteria with anti-inflammatory properties (i.e., Akkermansia) [110].

On the other hand, trying to affect gut microbiota by modifying it with prebiotics, probiotics, or symbiotics up to fecal transplantation could represent a promising pattern in obesity treatment [111].

Some postbiotics, particularly short-chain fatty acids (i.e., acetate, propionate, and butyrate), bacteriocins, and muramyl dipeptides, may help reduce appetite and food intake because of their ability to regulate metabolic hormones such as GLP-1 and PYY [108,112]. An RCT conducted in 2021 [113] demonstrated that supplementation with Saccharomyces boulardii and superoxide dismutase in obese individuals resulted in significant WL, primarily in terms of FM, while preserving LM. The composition of meals, interaction with the gut microbiota, and individual factors such as sex or degree of obesity influence GLP-1 secretion [112].

Dietary interventions, particularly those targeting the relationship between macronutrients and microbiome, can affect postprandial GLP-1 secretion but cannot replace pharmacological treatments [114].

Further studies could evaluate the combination of GLP1Ras and therapy affecting microbiota to contribute to the durability of anti-obesity medication effects.

### 3.5. PA in Obese People Treated with Incretin-Based Medications

Lifestyle interventions (including EXE), remain the first-line approach for achieving and maintaining healthy weight loss. Pharmacological treatment may be considered as an adjunct when lifestyle changes alone are insufficient, but they do not replace the fundamental role of lifestyle modification. These approaches support muscle preservation and overall metabolic health, reflecting the critical role of mitochondrial ATP production in converting nutrients into cellular energy necessary for muscle function, brain activity, and metabolic processes [115]. Despite the increasing use of new FDA-approved anti-obesity medications, PA remains essential in the comprehensive management of obesity [116] (Table 2).

PA, defined by the World Health Organization (WHO) as “*any bodily movement produced by skeletal muscles that requires energy expenditure*” [117], and EXE, a subgroup of PA, characterized by a plan, a repetitive and structured program [109], are usually not sufficient alone to promote meaningful WL. Still, they are among the most critical factors in fostering the long-term maintenance of WL [118]. However, the beneficial effects of PA and EXE on metabolic and mental health are well established [119].

To prevent weight gain and achieve health improvements, the WHO [120] and the American College of Sports Medicine [118] guidelines recommend five or more days per week of moderate-intensity PA (150–250 min/week), plus almost two days/week of strength EXE. Nevertheless, 150–250 min/week of moderate-intensity PA results in modest weight reduction. Instead, WL maintenance may require greater amounts of PA (>250 min/week) [118]. Strength training and other strategies to preserve lean body mass are likely to be particularly important in patients using AOMs, as well as in older adults who may have SO [121,122]. In these patients, excessive loss of LM, particularly muscle mass, is crucial because it increases the risk of frailty, falls, and fractures [123,124].

Some authors have shown that AOMs are more effective in WL than the lifestyle approach alone [62,125]; for example, Wadden et al. reported that WL induced by lifestyle modification is approximately 5–10% [62]. In another study by Rubino et al. [21], participants were assigned to receive either semaglutide, liraglutide, or placebo, along with counseling for diet reduction (−500 kcal/day) and PA (≥150 min/week). Semaglutide, added to counseling for diet and PA, resulted in an average WL of 15.8% compared to 6.4% with liraglutide.

**Table 1 nutrients-18-00131-t001:** Evidence of nutritional interventions or supplementation.

Authors	Sample	Intervention Modalities	Result
Sofi F et al. [71]	N: 118; Mean age: 51.1 yrs Mean BMI = 30.6 ± 4.9 (kg/m^2^)	2 different diets: vegetarian or Mediterranean (randomized, open, crossover dietary trial) Duration: 2 intervention periods, each lasting 3 months.	After 6 months, no differences between the 2 diets in body weight were observed (vegetarian: −1.88 kg, Mediterranean: −1.77 kg). Vegetarian diet reduced more LDL (9.10 mg/dL, *p* = 0.00); Mediterranean triglycerides (12.70 mg/dL, *p* < 0.01). Vegetarian diet was associated with vitamin B12 reduction (32.32 pg/m, *p* < 0.01)
Sacks F et al. [73]	N: 811 Mean age: 51 ± 9 yrs Mean BMI = 33 ± 4 (kg/m^2^)	4 diets with different percentages of protein, fat, and carbohydrates, respectively: 20, 15, and 65%; 20, 25, and 55%; 40, 15, and 45%; and 40, 25, and 35%. Duration: 2 years	Weight loss:4 kg. It was similar in participants assigned to a diet with 25% protein or 15% protein (3.6 and 3.0 kg, respectively; *p* = 0.22); the same in those assigned to a diet with 40% or 20% fat (3.3 kg, *p* = 0.94); there was no effect on weight loss of carbohydrate level through the target range of 35 to 65%. Waist circumference change did not differ sinificantly among the diet groups.
Shai I et al. [77]	N: 322 Age: 40–65 yrs BMI ≥ 27 (kg/m^2^)	3 different diets: low-fat, restricted-calorie; Mediterranean, restricted-calorie; or low-carbohydrate, non-restricted-calorie. Duration: 2 years	Weight loss:−2.9 kg for the low-fat group, −4.4 kg for the Mediterranean-diet group, and −4.7 kg for the low-carbohydrate group. Changes in fasting plasma glucose and insulin levels were more favorable for Mediterranean than low-fat diet.
McManus K et al. [78]	N: 101 Mean age: 44 ± 10 yrs Mean BMI = 33.5 ± 4 (kg/m^2^)	2 different diets: moderate-fat diet (35% of energy) or low-fat diet (20% of energy). Duration: 18 months	Moderate-fat diet induced mean reduction in weight of 4.1 kg, in body mass index of 1.6 kg/m^2^, and in waist circumference of 6.9 cm, whereas low fat diet increased body weight of 2.9 kg, BMI of 1.4 kg/m^2^ and waist circumference of 2.6 cm, respectively (*p* < or =0.001).
Verreijen AM et al., [80]	N: 100; Mean age: 62.4 ± 5.4 yrs Mean BMI = 32 ± 4 (kg/m^2^)	2 different hypocaloric diets, normal or high protein with or without a program of resistance training Duration: 10 weeks	High protein diet and exercise did not significantly affect change in body weight, FFM and fat mass; high protein in combination with exercise significantly increased FFM (+0.6 ± 1.3 kg, *p* = 0.011).
Miketinas, D. C. et al. [81]	N: 345; Mean age: 52.5 ± 8.7 yrs Mean BMI = 32.6 ± 3.9 (kg/m^2^)	4 low-calorie diets with different macronutrients and a dietary fiber intake ≥20 g/day Duration: 6 months	Body weight (kg ± SD) decreased by −5.8 ± 5.0, −5.8 ± 4.9, −7.1 ± 4.9, and −10.3 ± 6.3 kg for each type of diet, respectively.
Brunani et al. [86]	N: 40; Mean age: 52.55 ± 5.06 yrs Mean BMI = 44.45 ± 6.54 (kg/m^2^)	Rehabilitation with a low-calorie diet + structured PA. 4 groups: (1) control, (2) protein supplementation, (3) BCAA supplementation, and (4) EAA mixture with tricarboxylic acid cycle intermediates supplementation Duration: 4 weeks	Similar weight loss in all groups; the group with amino acid mixture had an increase in lean mass (≈+2.8 kg), but the BCAA and protein groups showed no significant differences compared to the control group.
Kang M. et al. [90]	N: 64; Mean age: 63.17 ± 4.53 yrs Mean BMI = 24.15 ± 2.69 (kg/m^2^)	Consumption of ONS tests with protein including BCAAs and ONS placebo with the same calories but without protein Duration: 12 weeks	Lean body mass: −0.47% in placebo group and +0.26 in test group. Body fat mass: +0.52% in placebo group and −0.25 in test group. No difference in muscular strength, physical performance, or inflammatory markers.
Camajani E. et al. [91]	N: 16; Mean age: 60 yrs Mean BMI = 37.6 ± 4.4 (kg/m^2^)	LCD (1000 kcal/day) with supplementation of whey protein, L-leucine (4.1 g), and vitamin D vs. control (without supplementation) Duration: 45 days	The supplemented group preserved lean mass significantly more than the control group. Total weight loss was similar between groups (−4.6%). A significant reduction in BMI (37.6 vs. 35.7 kg/m^2^), and waist circumference (107 vs. 102.4 cm), was observed in all patients.
Nederveen J.P. et al. [105]	N: 65; Mean age: 26 ± 1 yrs Mean BMI = 30.5 ± 0.6 (kg/m^2^)	Multi-ingredient supplement (forskolin, green coffee bean extract, green tea extract, beet root extract, α-lipoic acid, vitamin E, and Coenzyme Q10) vs. placebo Duration: 12 weeks	The intervention group showed a decrease in body weight (Δ = −2.2 ± 2.8 kg) and a reduction in fat mass (Δ = −1.4 ± 2.5 kg). No changes in total fat-free mass were observed in either treatment arms.
Bobe G. et al. [106]	N: 81 Mean age: 39 ± 9 yrs Mean BMI = 34.6 ± 6.1 (kg/m^2^)	600 mg/d of (R)-α-lipoic acid (R-LA) supplementation vs. placebo Duration: 24 weeks	The R-LA group had a greater BMI reduction than the placebo at 24 weeks (−0.8; *p* = 0.04). Women and those with obesity (BMI ≥ 35) experienced more significant weight loss (−5.0% and −4.8%; *p* < 0.001) and body fat reductions (−9.4% and −8.6%; *p* < 0.005).
Rondanelli M. et al. [113]	N: 25; Mean age: 53.68 ± 9.71 yrs Mean BMI = 34.83 ± 3.05 (kg/m^2^)	Low-calorie diet with the addition of 250 mg of Saccharomyces boulardii and 500 IU of Superoxide Dismutase (in gastro-resistant capsules, twice a day). Duration: 2 months	The intervention group showed a decrease in body weight (Δ = −2.73 kg) and fat mass (Δ = −3.13 kg), especially at the visceral level. Both groups showed no changes in lean mass.

Legend: ONS = oral nutritional supplementation. Data are shown as mean ± standard deviation.

Lundgren et al. [125], in a RCT involving 195 adults with obesity, studied four groups: (1) EXE plus placebo, (2) liraglutide plus usual activity, (3) combination of EXE and liraglutide, (4) placebo plus usual activity. The 52-week EXE program provided two sessions per week, each lasting 45 min, comprising 30 min of vigorous-intensity indoor cycling followed by 15 min of circuit training. After one year, the combination group resulted in an average WL of 9.5 kg, superior to that of both EXE alone and the liraglutide group (4.1 kg and 6.8 kg, respectively). The combination strategy also significantly reduced body fat percentage and improved metabolic health markers, including insulin sensitivity and cardiorespiratory fitness. In an extension of Lundgren’s study, Jensen et al. [126] reported that within the year following the cessation of treatment, weight regain was 6.0 kg larger after the discontinuation of liraglutide compared to the cessation of supervised EXE, and 2.5 kg [−1.5 to 6.5] greater compared to the termination of combination treatment. Higher proportions of participants achieved significant WL thresholds: 70.9% lost ≥10%, 55.6% lost ≥15%, and 38.5% lost ≥20% with semaglutide, compared to 25.6%, 12.0%, and 6.0% with liraglutide, respectively (all *p* < 0.001) [126].

The effects of GLP-1 RAs on weight reduction are dose- and duration-dependent [127]. Both the STEP-4 [128] and SURMOUNT-4 [129] studies showed that after 20 weeks of pharmacological treatment with semaglutide and after 36 weeks with tirzepatide, respectively, people with obesity tended to regain weight if the pharmacological treatment was stopped. The economic justification for drug reimbursement is weak for most patients, particularly those with a lower BMI [130]. This has led to emerging sustainability problems that need to be addressed, along with other obstacles (such as their state of health and lack of time), to engaging in an active lifestyle, as referred to by patients suffering from chronic Noncommunicable Diseases (NCDs) [131]. On the other hand, although some people referred to PA as expensive [132], regular PA is cheaper than a pharmacological approach.

Many authors have focused on the effects of different EXE intervention strategies. Åkerström et al. [133] observed a positive impact of endurance training on β-cell sensitivity to GLP-1 in 24 female participants randomized into two groups: one performing 10 weeks of intermittent training (27 min) and the other performing continuous training (40 min). Knudsen [134] et al. observed that a 1 h treadmill walk at 65% of peak heart rate in the morning increased beta-cell sensitivity to GLP-1.

Mehrtash et al. [135] suggested that aerobic EXE (starting from 10 min to building 150 min/week) plus muscular strength EXE (2/3 times a week, for 30 min) are recommended to people treated with incretin-based medications. Furthermore, balance and mobility training, particularly strength training [122], is suggested for patients using AOMs [136].

Finally, the benefits of EXE extend beyond WL alone. Enhanced levels of cardiorespiratory fitness are associated with health advantages that are either independent of or augment the effects of WL [54,137] and physical function. An illustration of this is the observed improvement in the KCCQ-CSS and 6 min walking test in the previously mentioned STEP-HFpEF trial involving semaglutide [25].

**Table 2 nutrients-18-00131-t002:** PA and EXE evidence.

Authors	Sample	Intervention Modalities	Result
Rubino M et al. [21]	N: 338; Mean age: 49 ± 13 yrs Mean BMI = 37.5 ± 6.8 (kg/m^2^)	3 Groups: (1) Weekly semaglutide injections (2.4 mg, 16-week dosage increase) vs. placebo. (2) Daily liraglutide injections (3.0 mg, 4-week dosage increase) vs. placebo. (3) Placebo groups combined. (1); (2); (3) + counseling for diet and PA.	Semaglutide—15.8% vs. liraglutide—6.4% of weight (*p* < 0.001). Participants on semaglutide were more likely to lose 10% (70.9% vs. 25.6%), 15% (55.6% vs. 12.0%), or 20% (38.5% vs. 6.0%) of their weight compared to those on liraglutide (all differences being statistically significant, *p* < 0.001).
Lundgren JR et al. [125]	N: 195; Mean age: 42 ± 12 yrs Mean BMI = 37.0 ± 2.9 (kg/m^2^)	8 weeks: low-calorie diet. 4 Groups: (1) exercise plus placebo; (2) liraglutide plus usual activity; (3) combination of exercise and liraglutide; (4) placebo plus usual activity. EXE program: 52 weeks (6-week ramp-up). 45 min, twice a week, comprising 30 min of vigorous-intensity indoor cycling followed by 15 min of circuit training. Duration: 1 year	All active treatment groups lost more weight than the placebo group: the exercise group lost 4.1 kg, the liraglutide group lost 6.8 kg, and the combination group lost 9.5 kg. The combination group lost 5.4 kg more than the exercise group and 2.7 kg more than the liraglutide group. The combination strategy also reduced body fat by 3.9%, compared to the exercise group’s (1.7%) and the liraglutide group’s (1.9%)
Jensen et al. [126]	N: 109 (among participants in Lundgren’s study); Mean age: 43 ± 12 yrs Mean BMI = 32.6 ± 2.9 (kg/m^2^)	Follow up observation (one year after Lundgren’s planned intervention) Duration: 1 year	Participants regained an average of 6.0 kg more weight after stopping liraglutide than after stopping supervised exercise, and 2.5 kg more than after combination treatment. Over one year following the cessation of combined exercise and liraglutide, participants had reduced body weight (−5.1 kg; *p* = 0.040). and body-fat (−2.3%; *p* = 0.026) compared to after stopping liraglutide alone.
Rubino M et al. [128]	N: 902; Mean age: 46 ± 12 yrs Mean BMI = 38.4 ± 6.9 (kg/m^2^)	Participants received weekly subcutaneous semaglutide during the run-in phase. After 20 weeks, 89% of participants at the 2.4 mg/wk maintenance dose were randomized (2:1) to 48 weeks of continued semaglutide (n = 535) or placebo (n = 268), with both groups receiving lifestyle intervention. Duration: 20 + 48 weeks	Participants continuing with semaglutide lost an average of 7.9% body weight by week 68, while the placebo group gained 6.9%, marking a significant difference of 14.8% (*p* < 0.001). The semaglutide group also experienced reductions in waist size (9.7 cm), blood pressure (3.9 mm Hg), and improved physical functioning (2.5 points on SF-36) compared to placebo (all *p* < 0.001).
Aronne LJ et al. [129]	N: 783; Mean age: 48 ± 13 yrs Mean BMI = 38.4 ± 6.6 (kg/m^2^)	Participants received weekly subcutaneous injections of tirzepatide at either 10 or 15 mg for 36 weeks. After that period, 670 participants were randomly assigned to either continue tirzepatide (335 participants) or switch to a placebo (335 participants) for 52 weeks. Participants received lifestyle counseling to encourage adherence to a deficit diet (500 kcal/d) and at least 150 min of PA per week. Duration: 36 + 52 weeks	Participants who completed a 36-week lead-in lost an average of 20.9% of their weight. From weeks 36 to 88, weight change was −5.5% for tirzepatide compared to −14.0% for placebo (difference −19.4%, *p* < 0.001). By week 88, 89.5% of tirzepatide participants maintained at least 80% of their weight loss, versus 16.6% for placebo (*p* < 0.001). Overall, the mean weight reduction was 25.3% for tirzepatide and 9.9% for placebo.

Legend: EXE = exercise; PA = physical activity. Data are showed as mean ± standard deviation.

### 3.6. Brief Overview of Other Incretin-Based Medications on the Way

Within the class of GLP-1 RAs, oral formulations have emerged, with additional agents anticipated in the near future. Oral semaglutide is employed for the treatment of T2DM at daily doses of 3, 7, and 14 mg, demonstrating WL effects comparable to its injectable counterpart [138]. Two RCTs, OASIS-1 and PIONEER PLUS, evaluated oral semaglutide 50 mg daily in obese individuals without T2DM and with T2DM, respectively, achieving 17.4% WL versus 1.8% with placebo in those without T2DM and 9.8% WL versus 5.4% WL with oral semaglutide 14 mg in those with T2DM [139,140]. More recently, oral semaglutide 25 mg has been shown to induce a 13.6% WL after 71 weeks in obese individuals with T2DM [141].

Other oral agents under investigation include non-peptidic GLP-1 RAs (small molecules) that bind the GLP-1 receptor in a manner distinct from native GLP-1. Among these, orforglipron is in advanced development for obesity and T2DM. In the phase 3 ATTAIN-1 trial, it produced up to 11.2% WL with the highest dose of 36 mg daily in individuals without T2DM [142]. The effects observed in this population exceeded those reported in individuals with T2DM in a separate trial [143], and additional phase 3 RCTs are ongoing to further assess its safety and long-term benefits.

Injectable agents that combine GLP-1 RAs with compounds targeting other pathways are also advancing. Cagrisema, a combination of semaglutide and cagrlintide, each administered at 2.4 mg weekly, is currently under investigation. Cagrlintide is a long-acting analog of amylin, a hormone co-secreted with insulin by beta-cells, involved in glucose homeostasis by suppressing glucagon secretion, slowing gastring emptying, and promoting satiety through central nervous pathways [144]. The results from REDEFINE-1 and REDEFINE-2, demonstrated the efficacy of cagrisema, both in obese people with T2DM and those without T2DM [144,145]. Notably, in the latter population, cagrisema induced WL of up to 22.7%, significantly surpassing the effects of either component alone. Similarly, maridebart carfraglutide, known as MariTide, a long-acting peptide-antibody conjugate combining GLP-1 receptor agonism with GIP receptor antagonism, achieved 12.3–16.2% WL after 52 weeks in obese participants with slightly lower reductions in individuals with T2DM [146].

Finally, retatrutide represents the first triple agonist targeting GLP-1, GIP, and glucagon receptors. In fact, glucagon receptor agonism has been shown to possess significant anti-obesity effects, as evidenced by its strong reduction in food intake and increase in systemic energy expenditure [147]. In a phase 2 RCT, weekly doses of 12 mg of retatrutide induced over 24% WL in adults with obesity [148]. A phase 3 program (TRIUMPH) is currently underway to assess the safety and efficacy of retatrutide across multiple obesity-related conditions, including OSA, T2DM, CV disease, and osteoarthritis.

The principal findings from the key studies evaluating incretin-based agents discussed throughout this manuscript are summarized in Table 3.

The significant portion of WL associated with these medications will require a deeper understanding of the change in body tissue composition and eating habits, in order to adequately support sustained WL and preserve nutritional deficiencies and LM.

## 4. Discussion

AOMs cause rapid and significant weight loss, higher and faster than traditional weight loss by diet and exercise, and offer a real and valid alternative to BS [66]. The introduction of incretin-based medications had undeniably transformed the management of obesity, coinciding with a shift in the obesity paradigm. These drugs not only facilitate WL but also significantly affect various organ damage and enhance QoL. The weight reduction achieved through these medications is comparable to that resulting from invasive surgical procedures, which are often associated with long-term complications.

However, it is crucial to emphasize the necessity of continuous and long-term follow-up when treating people with these agents [67]. While GI AEs induced by GLP-1 RAs were already known from their decade-long use in the management of T2DM, the loss of LM warrants greater attention due to its potential clinical implications and the interventions required to counteract it, which remain inadequately understood. Regarding these two important, potentially interconnected aspects of the safety profile of incretin-based therapies, it is noteworthy that tirzepatide exhibits a safety profile comparable to that of GLP-1 RAs. Notably, data from the FDA Adverse Event Reporting System indicate a lower incidence of nausea and vomiting, potentially due to GIP receptor-mediated neuronal inhibition in the area postrema [149,150], whereas constipation appears more frequent, with the underlying mechanisms yet to be fully elucidated [151].

The importance of combining dietary interventions and structured PA to counterbalance lean mass loss during the weight loss process resulted from BS. The quality and type of diet during drug treatment with AOMs are crucial for the patient’s nutritional status, especially in the long term [67]. Although the benefits of the incretin-based medications in terms of WL and improved metabolic parameters are significant, their influence on eating patterns and dietary quality has not been widely reported, and an unbalanced nutritional intake accompanied by a calorie deficit can lead to macro/micronutrient deficiencies and loss of LM [59,63,73,152]. Conversely, this can decrease the likelihood of successful long-term weight reduction maintenance, especially when behavioral changes do not occur. Hunger relief can prevent the intake of essential nutrients, and increased fatigue, decreased activity tolerance, and reduced strength can affect regular EXE. Therefore, given the increasing use of these drugs, it is essential to take a multidisciplinary approach that includes personalized nutritional counseling to support patients in making healthy and balanced food choices [76,153].

Strategies such as the MD and the DASH diet appear to be more sustainable [77] in the long term and have fewer gastrointestinal side effects that often lead to malnutrition. The right protein intake plays a crucial role in preserving muscle mass loss (approximately 1.5 g/kg of high biological value protein), supported by the possible use of specific supplements, such as amino acids, vitamin D, B vitamins, and minerals [62,68,74,152]. Finally, growing interest in the role of gut microbiota and postbiotics in regulating metabolism and appetite is opening new therapeutic scenarios, although further scientific confirmation is needed [113].

Although it notes the difficulty of people with NCDs to achieve the minimum level of weekly PA necessary to obtain health improvement, regular PA and, in particular, EXE remain a key part of treating obesity [116]. Moreover, they are the most sustainable strategy to counteract SO and the LM loss caused by AOMSs [121,122]. Finally, they are crucial to maintain healthy physical function and fit CRF levels useful to WL [54,137].

Among the efficacious PA strategies, some authors suggested that mixed (aerobic + strength EXE) [135], continuous and intermittent endurance training [133,134], balance and mobility EXE [136] are useful to people taking AOMs.

Standard treatments for improving SMM and strength in obese and/or older patients, such as dietary protein intake and resistance EXE training, are often challenging; therefore, new pharmacological molecules are under development. Bimagrumab is a monoclonal antibody that binds to the activin type II receptor. Phase 1 and 2 trials showed that bimagrumab treatment significantly increased FFM and was also effective in reducing FM [154]. An ongoing trial is investigating the efficacy of bimagrumab alone or in addition to semaglutide to assess efficacy and safety in overweight or obese men and women [155]. Other monoclonal antibodies, trevogrumab (a myostatin signal inhibitor) and garetosmab (an activin signal inhibitor), are under development and should be, in the future, a valid strategy, in combination with supplements and training, to enhance fat loss and preserve muscle mass [156].

The results of currently available AOMs and of the others in development in terms of EWL% are comparable to those achieved by BS, which could represent a model to better understand the importance of a strict follow-up of patients during weight loss, paying attention to counterbalance FFM/LM loss with a particular attention to nutrition, both qualitatively and quantitatively, and structured PA. Weight regain, often occurring after BS, could also be experienced in people treated with AOMs. Knowing the main determinants of post-bariatric WR could help teams involved in the treatment of obesity engage people to prevent it from taking care of changing their lifestyle definitively.

In conclusion, when managing obesity with AOMs, healthcare professionals should assess nutritional status and the risk of malnutrition and provide personalized dietary counseling [61]. The therapeutic plan should include lifestyle modifications and a balanced, nutrient-rich diet, with the possible inclusion of fortified foods or supplements to prevent or correct nutritional deficiencies, avoid side effects [75,76], and provide an ongoing assessment to support WL maintenance over time, as is usually performed after BS.

## 5. Conclusions

The incretin-based medications have demonstrated considerable efficacy in facilitating WL; however, these agents require time to manifest their full effects. A significant portion of WL associated with these medications is attributed to the reduction in LM, which poses a particular concern for individuals with obesity, especially among older adults.

Healthcare providers must assess patients’ nutritional status and potential risk of malnutrition when administering AOMs. Providers should offer personalized dietary guidance, emphasizing lifestyle modifications and a balanced diet. This may involve the inclusion of nutrient-rich foods or supplements to address any nutritional deficiencies.

Furthermore, incorporating supervised EXE into the AOMs treatment regimen is beneficial for sustaining WL over the long term. Also, EXE appears to be beneficial in stabilizing WL following the discontinuation of pharmacological treatment.

In conclusion, the integration of comprehensive nutritional support and regular EXE, based on guidelines, is essential for individuals undergoing treatment with incretin-based medications for the care of obesity and to optimize long-term health outcomes.

## Figures and Tables

**Figure 1 nutrients-18-00131-f001:**
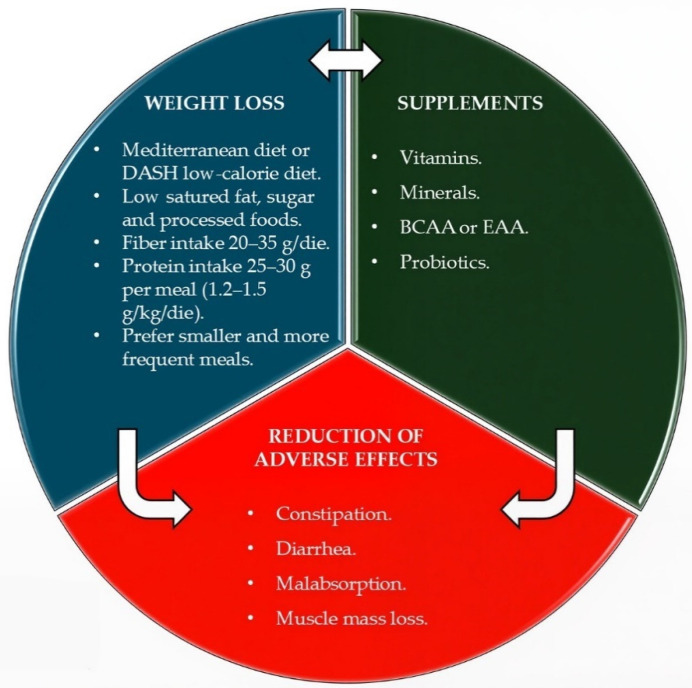
Summary of nutritional evidence.

**Table 3 nutrients-18-00131-t003:** Evidence from the main clinical trials on the use of incretin-based agents in people with obesity.

Authors	Sample	Intervention Modalities (Weeks)	Mean Weight at Baseline (kg)	Results
Pi-Sunyer X et al. [18]	N: 3731; Mean age: 45.1 ± 12 yrs Mean BMI = 38.3 (kg/m^2^)	Liraglutide 3.0 mg sc daily vs. Placebo; Duration: 56 weeks	106.2	Weight reduction (%): −8 vs. −2.6
Wilding JPH et al. [20]	N = 1961 Mean age: 46 yrs Mean BMI = 37.9	Semaglutide 2.4 mg sc weekly vs. Placebo; Duration: 68 weeks	105.3	Weight reduction (%): −14.9 vs. −2.4
Jastreboff AM et al. [22]	N: 2539; Mean age: 44.9 ± 12.5 yrs Mean BMI = 38	Tirzepatide 15 mg sc weekly vs. Placebo; Duration: 72 weeks	104.8	Weight reduction (%): −20.9 vs. −3.1
Knop FK et al. [139]	N: 709; Mean age:50 ± 13 yrs Mean BMI = 37·5 ± 6·5	Semaglutide 50 mg os daily vs. Placebo; Duration: 68 weeks	105.4	Weight reduction (%): −15.1 vs. 2.4
Wharton S et al. [141]	N: 205; Mean age: 47.5 ± 13 yrs Mean BMI = 36.0	Semaglutide 25 mg os daily vs. Placebo; Duration: 71 weeks	105.9	Weight reduction (%): −13.6 vs. −2.2
Wharton S et al. [142]	N: 3127; Mean age: 45.1 ± 11.9 Mean BMI = 37.0 ± 6.5	Orforglipron 36 mg os daily vs. Placebo; Duration: 72 weeks	103.2	Weight reduction (%): −11.2 vs. −3.1
Garvey TW et al. [144]	N: 3417; Mean age: 47.0 ± 11.8 yrs Mean BMI = 37.9	CagriSema 2.4 mg + 2.4 mg sc weekly vs. Placebo, Semaglutide 2.4 mg sc, Cargilintide 2.4 mg sc Duration: 68 weeks	106.9	Weight reduction (%): −20.4 vs. −3.0 (Placebo)
Jastreboff AM et al. [146]	N: 592; Mean age 51.5 ± 11.8 Mean BMI = 37.9	Maridebart cafraglutide 420 mg sc monthly vs. Placebo; Duration: 52 weeks	107.4	Weight reduction (%): −16.2 vs. 2.5
Jastreboff AM et al. [148]	N: 338; Mean age: 48.2 ± 12.7 yrs Mean BMI = 37.3	Retatrutide 12 mg sc weekly vs. Placebo; Duration: 48 weeks	107.7	Weight reduction (%): −24.2 vs. −3.1

Data are showed as mean ± standard deviation.

## Data Availability

Data sharing is not applicable.

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
