# Peer review of "Nutrition and Physical Activity in Optimizing Weight Loss and Lean Mass Preservation in the Incretin-Based Medications Era: A Narrative Review"

_nutrients, 2025, doi:10.3390/nu18010131_

Round 1

Reviewer 1 Report

Comments and Suggestions for Authors

Multi Digital Publishing Institute (MDPI): mdpi-nutrients-3974076-v1

Review

Title: Nutritional and Physical Strategies to Preserve Lean Mass in Obese Adults Treated with Incretin-Based Medications: A Narrative Review

This is a review of nutritional and physical strategies to preserve lean mass on incretin-based treatments.

Overall comments:

The topic in this review is a very important one for society.  The authors noted this as “narrative review, but may be more suitable to categorize it as systematic review?

Narrative reviews have looser categorization with more input from the authors though each journal is slightly different.  The authors went through different studies with their names so more suitable as systematic review.  For a narrative review, summaries of various studies would be more succinct.

Table 1 and Table 2 are very helpful, but as a narrative review, more interpretations of the authors should be stated, and these sections should be the main point of the manuscript in my view.

Recommend reorganizing some subsections, such as 2.2 should be the last section since this section describes agents which may be available in the future.

The aim of this review is similar to what was done before:

Neeland IJ, Linge J, Birkenfeld AL. Changes in lean body mass with glucagon-like peptide-1-based therapies and mitigation strategies. Diabetes Obes Metab. 2024 Sep;26 Suppl 4:16-27. doi: 10.1111/dom.15728. Epub 2024 Jun 27. PMID: 38937282.

Specific comments:

  1. Introduction

Lines 47: Replacing “normal mass” with another word.  What we label “normal” is somewhat artificial and using another word as “optimal weight” or “optimal mass” may be desirable, meaning that the optimal weight to maintain the homeostasis.

Lines 66-70: Recommend some editing of this paragraph.  The first sentence and the second sentence do not connect well since the first sentence seems to elaborate more on GLP-1RA (efficacy and safety profile), but the second sentence refers to a new combination type of medication (GLP-1RA + GIP), not necessarily related to efficacy and safety of GLP-1RAs.  Thus, “The first of these” is not suitable.

“A new combination which can act simultaneously on … is also underdevelopment for enhancing the effect of enhancing incretin-based therapy, first of which is tripeptide.

Line 71: Recommend considering replacing “Consequently” with another connecting word.

“Following the success of these medications, significant attention has shifted toward the changes in body composition

Line 74: Recommend inserting, “which” after sarcopenic obesity to avoid the sentence to be run-on.

“…sarcopenic obesity (SO), which is a condition affecting approximately 42-43% of individuals with obesity and is often associated with adverse clinical events.”

Line 77: Please avoid the indentation.

Line 78: Recommend replacing words, “specific structures within” since “structures” are not the ones regulate.  “Molecular signals” or “hormones” being secreted from the neuroendocrine systems…

Alternatively, deleting “by specific structures” may be enough.

Example:

“Food intake is finely regulated by the neuroendocrine systems at both central and peripheral levels where the actions of novel pharmacologic agents are thought to exert their effects.” (This still requires further investigation but at least recommend adding a reference here since it is an important statement).

There are additional organs which are reported to express GLP-1 receptors as below.

Glucagon-like peptide-1 (GLP-1) receptors are primarily located in the following tissues: Gastrointestinal tract: small intestine and stomach.  Brain: hypothalamus and brainstem.  Pancreas: beta cells.  Other tissues: heart, liver, and kidneys. 

Zhao X, Wang M, Wen Z, Lu Z, Cui L, Fu C, Xue H, Liu Y, Zhang Y. GLP-1 Receptor Agonists: Beyond Their Pancreatic Effects. Front Endocrinol (Lausanne). 2021 Aug 23;12:721135. doi: 10.3389/fendo.2021.721135. PMID: 34497589; PMCID: PMC8419463.

Liu QK. Mechanisms of action and therapeutic applications of GLP-1 and dual GIP/GLP-1 receptor agonists. Front Endocrinol (Lausanne). 2024 Jul 24;15:1431292. doi: 10.3389/fendo.2024.1431292. PMID: 39114288; PMCID: PMC11304055.

There should be a sentence in which the authors’ hypothesis which aligns with the title of

Nutritional and Physical Strategies to Preserve Lean Mass in Obese Adults Treated with Incretin-Based Medications”, in the introduction.  Thus, it should include some underlying mechanisms how lean muscle loss occurs in individuals who have been treated with GLP-1RAs.

After reading the introduction, the main theme of this review was unclear.

  1. Materials and Methods

Recommend to inclyude some search on “lean muscle loss or preservation” with GLP-1RA therapy in their literature search.

  1. Results

Since this is a review manuscript, so recommend this section to be summarizing the findings from the clinical studies which align with the title.

Recommend summarizing this section so that each paragraph does not just explain about each study.  Table 1 and Table 2 should have enough information for each study so that this section should include some relevant findings regarding the main topic for the authors.

Recommend focusing on 2.3 and 2.4 since these sections seem to be more relevant to the topic on which the authors have tried to focus.

Recommend 2.2 to be written later since these are agents which are being investigated or future pharmaceutical agents.

Lines 236-246: Recommend describing Amylin since this is different from incretin-based therapy.

Line 334: Recommend replacing “reduce energy expenditure” with another words.  Do the authors mean that the use of AOM “reduces baseline energy requirements?”

“Basal metabolic rate?”

Alternatively, “reduces baseline energy availability?”

Please clarify.

It is also important to recognize that individuals probably had consumed unbalanced diets, becoming obese, thus, it may be reasonable to expect not nutritionally well with the use of AOM.  Furthermore, since they are monitored as they start AOM, these deficiencies are noted when they start AOM.  Especially, vitamin D is a fat-soluble vitamin, in addition to vitamins A, B, and E, so when fat-volume changes occur they are likely to be altered drastically.

Line 354: As mentioned above, recommend adding, “baseline” to “energy requirements” since depending on types of activities, energy requirements easily change.

Lines 397-400: Recommend clarifying this section.  This section does not need to be a separate paragraph since it pertains to fiber intake.  Furthermore, recommend replacing words, “malnutrition” and “reduce lean protein intake”.

It is unlikely that people who were obese and taking AOM are considered as having “malnutrition”.  They may have imbalance in nutritional intake.  Please clarify “reduce lean protein intake”.  Is this something individuals are intentionally doing or problem with absorption.  It is unclear from the sentence.  It is not surprising that when individuals are on AOM, there is a need for vitamin supplementation since nutritional homeostasis is being altered, and until the new and stable homeostasis is reached, there may be times when certain nutrients are deficient.

Line 402: Again, recommend replacing “malnutrition” with another word or phrase such as imbalance in nutritional intake or nutritional imbalance.

Malnutrition has more serious connotation and seems to connect with starvation.

Lines 402-403: As noted earlier, this sentence should be noted earlier, and some paragraphs can be condensed with a better summary. 

Lines 403-406: Recommend editing this sentence.

The second part is rather run-on.

“can contribute to prolonged fasting periods and insufficient dietary variety with inadequate protein intake, which can lead to micronutrient deficiency and a loss of muscle mass.  “Reduced energy expenditure” in this sense does not make sense.  Do the authors mean this is a bad consequence?  Depending on activity, energy expenditure changes rapidly.  A reduction in energy expenditure is not always a negative consequence.  A reduction in basal energy requirement which depends on how the body uses energy at baseline, but this is per what the body needs at rest, and a reduction does not necessarily imply a negative consequence.

Lines 417-464: This section has many great points which align well with the title.  Reorganizing the section can improve its readability.  Recommend splitting this section into 2-3 paragraphs may help the readers with better focus.

It may be helpful to describe why magnesium, zinc or other micronutrients and vitamins are required for the body.

What are known about magnesium with the use of GLP-1RA?

Lines 456-464: It may be important for the authors to describe these amino acids in detail for the readers.  There are somewhat different types of amino acids listed here.

It may be helpful for the readers to see a table with different macro- and micro- nutrients supplementations suggested.

Lines 465-468: Recommend this paragraph to be integrated into the paragraph above and reorganize them into several paragraphs concentrating on dietary supplements including micro-nutrients.

It may be helpful to have a special section on microbiota since there is an intimate relationship between GI and CNS (the gut-brain axis).

Lines 490-492: This first sentence seems not to align with our body system.

Regardless of the use of AOMs, lifestyle intervention should be central to weight loss.

If this is uncertain as the authors state, please provide a reference.

Individuals who become bedridden, lean mass loss is pronounced as we already know.

Balanced dietary intake and physical activity should be focused regardless of AOMs.

It may be helpful to think about biological systems and how nutrients are metabolized in our body.  We consume nutrients so that we produce cellular energy in the form of ATPs through biochemical pathways.  ATPs are essential for many cellular reactions and metabolism.  Where we generate most ATPs are in mitochondria and they are central to nutrient metabolism.  Mitochondrial energy is important in muscle movements and our brain activity as well.

This information may be beyond the content of this review.  However, it is always important to think about biological systems, beyond what we can physically observe. 

Recommend section 2.6 on Bariatric surgical therapy to be discussed earlier in the manuscript if this is to be included.  Please try to make this review align with the title.

Line 606: Recommend correcting “leucin-rich proteins” to “leucine-rich proteins”.

Recommend putting abbreviations in alphabetical order.  It is great that this was done, but the use of alphabetical order would make it easier to find what the readers are looking for.

The topic in this review is timely and very important.  However, it is important that the content aligns with the title.  Furthermore, it seems as though it is more of systematic review than narrative review.

Elaborating on the topic of macro- and micro- nutrient deficiencies, and what are known about each in detail, possibly in a table would be very informative for the readers.

Please also try to theorize the underlying biochemical pathways of nutrient metabolism if possible since they would be the center of nutrient homeostasis.

Thank you very much for allowing me to review this manuscript. 

Sincerely,

Reviewer 2 Report

Comments and Suggestions for Authors

The Abstract needs to be rewritten. The author should refer to the Abstract sections of other articles for guidance. The current abstract is of poor quality. I did not see what the purpose of this study is. Additionally, the main conclusions of the study are not clearly stated in the abstract. Finally, the abstract focuses excessively on the research background, which is unnecessary.

The subheadings in the manuscript are incorrect. Please review them.

Method and materials: What materials were used in the study?

The methods section is not detailed enough. How did you handle cases where it was unclear whether an article should be included?

Have you reviewed the reference list to supplement the literature?

Results: The results of many clinical studies are described in excessive detail. Please focus on content related to changes in obesity indicators.

Results: The content in this section lacks structure. Please further categorize these drugs and group those with common characteristics for discussion. The current approach seems to randomly select a drug and describe its weight loss effects, which is an inappropriate writing technique.

Section 2.3: The description of liraglutide's weight loss effects is overly detailed and unnecessary.

Nutritional supplementation is a very important part of this study. What nutritional recommendations do you have?

There is too little research on gut microbiota. Are there other related studies on gut microbiota? Please supplement.

Can you summarize any patterns in studies related to gut microbiota? For example, is supplementing with lactic acid bacteria or yeast more effective in improving obesity?

Does supplementing with nutrients that promote the growth of gut microbiota, such as dietary fiber, help improve obesity? Please gather relevant studies and discuss their patterns.

Round 2

Reviewer 1 Report

Comments and Suggestions for Authors

Multi Digital Publishing Institute (MDPI): mdpi-nutrients-3974076-v2

Review

Title: Nutritional and Physical Strategies to Support Weight Loss and Preserve Lean Mass in Obese Adults Treated with Incretin-Based Medications: A Narrative Review

Overall comments:

This is a review of nutritional and physical strategies to preserve lean mass on incretin-based treatments.

The topic in this review is a very important one for society. 

Although there seems to be some evidence that the authors have tried to improve the manuscript, it is still not well aligned with the title, unfortunately.  It is important that the review focuses on “Nutritional and Physical Strategies to Support Weight Loss and Preserve Lean Mass”. 

The abstract seems to be fairly well written, and does focus on what the title says.  However, this is not aligned in the main text.  A strategy to improve the review may be to use the skeleton noted in the abstract, and elaborate it in the main text, especially in the result section.

The current state of the result section is too long, and without a well-organized theme as in the abstract.

Specific comments:

  1. Introduction:

Lines 71-74: Recommend editing this sentence.

The authors are trying to state that in addition to contributing to improving QoL, and GLP-1RAs also help to minimize organ damages, correct?

Lines 66-74:

Lines 111-119:

The above two paragraphs are the main topic for the authors and they have to be discussed more extensively and placed in the main section of the introduction.

Since the authors are going into extraneous issues although they are important in the therapy with GLP-1RAs, the main topic should be elaborated so that the readers can understand what the authors are trying to convey through this narrative review.

Lines 75-86: This paragraph should be placed later since GLP-1R agonists and GIP combination medications are newer ones.

Recommend focusing on GLP-1RAs first and going into findings related to weight loss and lean mass loss. 

In the introduction, it is great to focus on the neurological effects of GLP-1RA; however, the focus on this review is on weight loss and lean mass changes so it would be more important to state what happens with the use of GLP-1RA on these factors more than how GLP-1RAs work through

  1. Methods:

For a narrative review, an article could be more free-styled.

Line 139: Recommend revising this sentence.

In preparation for this review, a search of the MEDLINE (Pubmed) was conducted …

  1. Results:

For a narrative review, an article could be more free-styled.

It is great to have subsections to focus on certain topics.  However, it is not enough to list the studies in a narrative review. 

  • Recommend revising the subtitle here.

“Weight reduction and organ damage ameliorations (?) in obese people”

The authors are trying to mention that there are beneficial effects seen beyond weight reduction, correct?

Lines 161-203: This section has a pretty good theme; however, better summarizing studies which support the subtitle would improve the review.  Especially the second and third sections align with what the authors wanted to state.  Recommend editing this section into two paragraphs maximum because this is not the main theme of the review.

  • Recommend mentioning about adipose tissue change or fat mass change here before mentioning about SMM and bone density. Or alternatively, change the subtitle to change in body tissue composition.

  • Loss of adipose tissue or fat mass (visceral vs subcutaneous etc.)
  • Change in skeletal muscle mass (quality)
  • Change in bone density (quality).

Bhandarkar A, Bhat S, Kapoor N. Effect of GLP-1 receptor agonists on body composition. Curr Opin Endocrinol Diabetes Obes. 2025 Dec 1;32(6):279-285. doi: 10.1097/MED.0000000000000934. Epub 2025 Oct 10. PMID: 41076575.

  • Recommend

3.4. Evidence of Nutritional Interventions, Supplementation, and PA in Obese People Treated 318 with BS

It is great to have a section on bariatric surgery, but this section is too long.

It just needs one paragraph for a summary or a comparison table.

This could just be a subsection of below because the main topic pertains to individuals treated with incretin-based medications.

Recommend aligning with the title.

3.5. (3.3) Evidence of Nutritional Interventions or Supplementation in Obese People Treated with Incretin-Based Medications

This section is important and the authors have made some improvements.  However, it would be desirable to have a good description of what often happens with GLP-1RA treatment.  Then, providing remedies to make improvements as the authors listed.

The following 2 sections can be in GI related section, somewhat separate from nutrition deficiencies, even though they are related.

GI side effects should be in a different section, perhaps prior to starting the post-treatment nutritional changes.  This section should focus on nutritional changes and needs.

Microbiota is also important, but this section should really focus on nutrition.

It is important to have things well organized so that it is easier for the readers to follow the content of this review. 

Line 396: “a reduction in basal energy requirement” is not a bad thing.  If one’s weight is reduced, it is not surprising that required basal energy is reduced if this reduction is proportional to the body weight.

Line 422: Basal energetic needs should be reduced, and it is not a bad thing.

Basal energetic needs, or basal metabolic rate (BMR), are the minimum calories your body burns at rest for essential functions like breathing, circulation, and cell maintenance, making up 50-80% of total daily energy use. These needs vary by age, sex, weight, body composition (muscle burns more), and health status, often estimated using formulas like the Harris-Benedict equation, and are crucial for understanding overall calorie requirements for weight management. 

With a loss of weight, BMR should not be the same as prior to treatment so a reduction is a normal process, not abnormal.

When deficiencies of micro- or macro- nutrients occur, this is not desirable.

Recommend finding additional articles which studied micro- nutrient deficiencies pre- and post- GLP-1RA treatment.

Vitamin D, B vitamins (especially B12, folate), Iron, Calcium, Zinc, and Vitamins A, C, E.

Mozaffarian D, Agarwal M, Aggarwal M, Alexander L, Apovian CM, Bindlish S, Bonnet J, Butsch WS, Christensen S, Gianos E, Gulati M, Gupta A, Horn D, Kane RM, Saluja J, Sannidhi D, Fatima Cody S, Callahan EA. Nutritional Priorities to Support GLP-1 Therapy for Obesity: A Joint Advisory From the American College of Lifestyle Medicine, the American Society for Nutrition, the Obesity Medicine Association, and the Obesity Society. Am J Lifestyle Med. 2025 May 30:15598276251344827. doi: 10.1177/15598276251344827. Epub ahead of print. PMID: 40452753; PMCID: PMC12125019.

  • Dietary pattern recommendations
  • Diet recommendations with supplement to counter macro- and micro- deficiencies

3.6. PA in Obese People Treated With Incretin-Based Medications

Recommend shortening this section to focus on how exercising can counter skeletal muscle loss with GLP-1RA therapy, focusing on PA benefits on preserving lean mass, rather than the benefit of PA for additional weight loss although this is also important, but please focus on the title of this review.

(3.3 This section should be the last subsection as recommended in the earlier review.)

Recommend focusing on the title of this review.

Overall, much more streamlining would be recommended to improve the readability of this review.

Since the authors’ focus is on “Strategies to Support Weight Loss and Preserve Lean Mass in Obese Adults Treated with Incretin-Based Medications”, it is important that extraneous findings are minimized and focusing on their theme of this review. 

Thank you very much for allowing me to review this manuscript. 

Sincerely,

Reviewer 2 Report

Comments and Suggestions for Authors

Approved

Author Response

Thank you.

Round 3

Reviewer 1 Report

Comments and Suggestions for Authors

Multi Digital Publishing Institute (MDPI): mdpi-nutrients-3974076-v3

Review

Title: Nutritional and Physical Activity in Optimizing Weight Loss and Lean Mass Preservation in the Incretin-Based Medications Era: A Narrative Review

Overall comments:

Recommend reviewing the spelling especially in the title prior to submission.

Nutritio (n is missing).

It is important to correct minor errors prior to submission (especially the title).

Specific comments:

Abstract:

Line 25: This “narrative” review …

Line 27: Recommend editing here,

Methods: Literature review was conducted …, instead, deleting “A narrative review …”

Lines 41-44: Recommend editing this sentence (please re-read it).

While incretin-based medications produce substantial weight “loss”, their impact “on “lean mass” and “bone mass”? underscores the necessity of integrating … , and optimize long-term (health?) outcomes.”

  1. Introduction:

Line 57: Recommend replacing “contributing to obesity”, with “resulting in obesity” or “lead to obesity” since this is what happens with the abnormalities the authors noted in the sentence.

In introduction, there should be some type of statement about the goals set forth by the authors in this “narrative” review in the last paragraph of the introduction instead of the description of pipeline medications.

The introduction (https://www.cwauthors.com/article/How-to-write-an-introduction-to-an-academic-article)

“The introduction to an academic article is the first section of the paper, immediately following the abstract.

One of the most important functions of an introduction is to answer the question ‘why?’: why was the study performed, and why is it interesting and/or important? Given that the introduction is the beginning of the paper, it also serves to tell the reader why they should read the rest of the paper and prepares them to understand the importance and implications of the results.”

Lines 116-129: This section is more impactful if this is placed in the discussion section, not in the introduction.  “…remains to be fully elucidated” is not a good way to conclude the introduction section.

It has an effect to conclude the article, and the readers would stop reading at this point.

The last paragraph should be similar to what is written in the abstract, but should expand it.

“This narrative review addresses emerging clinical concerns with the use of ….”

  1. Methods:

Narrative Reviews involve looking at literature across a specific topic and “synthesizing what you have learned”. You can either look at one specific database, or across multiple databases (https://laneguides.stanford.edu/types-of-reviews/narrative).

Two paragraphs can be merged into one paragraph.

“In preparation for this review, a search of …….within the last ten years.  The search terms used were …..

The sentence, “Disagreements ….” can be added to the previous paragraph.

  1. Results:

Brignardello-Petersen R, Santesso N, Guyatt GH. Systematic reviews of the literature: an introduction to current methods. Am J Epidemiol. 2025 Feb 5;194(2):536-542. doi: 10.1093/aje/kwae232. PMID: 39038802; PMCID: PMC11815505.

This review is not exactly systematic review as per strict definition, but if this were a narrative review, it is important that the authors provide insightful synthesis and interpretation, not just making a list of grouped studies.

No need to make a study per paragraph.  It is more important to provide a streamline summary of studies related to the subtopic instead.  Otherwise, it is too difficult to read them through and grasp the contents.

Each subsection should be 3-4 paragraphs. 

Lines 272-281: Amylin related agents are not GLP-1RA agents.

3-1: the topic is important, but the section should be much shorter.

Recommend focusing basically on sections, 3-2 and 3-5.

Strategies sections should be important part of narrative review, and these sections should be central to this review with the interpretation provided by the authors.

3-3 should be placed later than other sections.

3-4 can be included and informative just as compared to GLP-1RA agents (not really important in the topic of this review.

Overall, still recommend making drastic structural reorganizations and streamlining so as to just focus on the topic of this narrative review.  Extra information may be informative but should not take up a large portion of this manuscript.

Since the authors’ focus is on “Nutrition and Physical Activity in Optimizing Weight Loss and Lean Mass Preservation in the Incretin-Based Medications Era”, it is important that extraneous topics are drastically minimized and focusing on their theme of this review.

Elaborating on the abstract theme is much better for this narrative review than how it is set up right now.

Thank you very much for allowing me to review this manuscript.

Sincerely,

Comments on the Quality of English Language

English sentences are fine, but recommend improving organizations and paragraphs.
